# Metabolomic and Transcriptomic Analyses Provide Insights into Metabolic Networks During Kiyomi Tangors Development and Ripening

**DOI:** 10.3390/plants14172751

**Published:** 2025-09-03

**Authors:** Xin Song, Tingting Wang, Peng Zhao, Yanjie Fan, Ligang He, Yu Zhang, Zhijing Wang, Xiaofang Ma, Cui Xiao, Yingchun Jiang, Fang Song, Liming Wu

**Affiliations:** 1Hubei Key Laboratory of Germplasm Innovation and Utilization of Fruit Trees, Institute of Fruit and Tea, Hubei Academy of Agricultural Science, Wuhan 430064, China; songxin@hbaas.com (X.S.);; 2National Key Laboratory for Germplasm Innovation & Utilization of Horticultural Crops, College of Horticulture & Forestry Sciences, Huazhong Agricultural University, Wuhan 430070, China; 3Hubei Hongshan Laboratory, Wuhan 430070, China

**Keywords:** transcriptome, metabolome, citric acid, sucrose, abscisic acid

## Abstract

Flavor formation in citrus fruit is governed by complex and dynamic changes in primary and secondary metabolism during development and ripening. Here, we integrated metabolomic, hormonal, and transcriptomic analyses to elucidate the regulatory landscape underlying soluble sugar and organic acid metabolism in ‘Kiyomi’ citrus fruit. A total of 1679 metabolites were identified, revealing stage-specific reprogramming of metabolic pathways, including a sharp decline in citric acid after 90 days after flowering (DAF) and peak accumulation of sucrose at 180 DAF. Hormonal profiling showed that abscisic acid (ABA) progressively accumulated from 120 to 210 DAF, while 1-aminocyclopropane-1-carboxylic acid (ACC) peaked at 120 DAF and declined thereafter, suggesting distinct and temporally coordinated roles in ripening regulation. Transcriptomic profiling uncovered widespread temporal shifts in gene expression, with the most pronounced changes occurring between 180 and 210 DAF. Co-expression network analysis identified gene modules associated with sugar and acid accumulation, and highlighted transcription factors from the ERF, MYB, NAC, and HSF families as candidate regulators of ripening-related metabolic transitions. These findings provide a comprehensive framework for understanding the coordinated molecular and metabolic programs underlying flavor development in non-climacteric citrus fruit and offer candidate genes for the genetic improvement of fruit quality.

## 1. Introduction

Fruit ripening is a highly complex and well-regulated biological process, involving significant changes in taste (sweetness and acidity), aroma, texture (softening and firmness), and appearance (color) [1]. Citrus fruits are typical hesperidia, with leathery peel surrounding the edible fruits portion, and have non-climacteric fruit ripening characteristics [2,3]. The development of citrus fruits can be divided into three stages: cell division stage, expansion stage, and ripening stage [4], and the dynamic balance between sugar accumulation and organic acid degradation plays a central role in defining the taste profile, directly affecting consumer acceptance and marketability [5]. Hence, knowledge of the molecular regulation of sugar and acid metabolic processes during fruit ripening is essential for genetic improvement and promotion of fruit quality in citrus.

Major phytohormones, including ethylene, abscisic acid (ABA), and auxin, serve as key regulators of metabolic pathways involved in fruit ripening. Auxin has been identified as a critical modulator of fruit set and development in species such as strawberry (*Vaccinium uliginosum*) and tomato (*Solanum lycopersicum*), primarily through tightly regulated spatial and temporal patterns of auxin distribution and signaling [6,7]. Ethylene plays a fundamental role in initiating and orchestrating ripening in climacteric fruits like tomato, banana (*Musa acuminata*), and apple (*Malus domestica*), with its regulatory mechanisms extensively characterized [8,9]. Additionally, ABA is recognized as the dominant hormone controlling ripening in non-climacteric fruits, with substantial increases in ABA levels observed during the ripening of citrus and strawberry [10]. Over-expression of *FaNCED1* gene in strawberry significantly evaluates the ABA content in fruits, and thus promotes fruit ripening, including sugar accumulation [11], anthocyanin biosynthesis [12], and eugenol production [13]. However, the mechanisms determining fruit ripening and quality by ABA in non-climacteric fruits remains to be elucidated.

The regulation of sugar and acid metabolism during fruit ripening involves complex networks of transcription factors (TFs) that modulate gene expression in metabolic pathways [14,15]. In apple, MdAREB2 (ABA-responsive element binding 2) improves sugar accumulation of fruits by transcriptional activating the sugar transport genes *MdSUT2* (*sucrose transporter 2*) and *MdTMT1* (*tonoplast monosaccharide transporter 1*) [16]. In grape (*Vitis vinifera*), VvWRKY22 has been shown to decrease the sucrose, glucose and fructose content by interacting with VvSnRK1.1/VvSnRK1.2 (sucrose non-fermenting-1-related protein kinase1.1/1.2) [17]. In pear (*Pyrus pyrifolia*), PpbZIP44 (basic leucine zipper 44) decreases citrate and malate, and increases fructose contents by directly regulating the expression of *PpSDH9* (*sorbitol dehydrogenase 9*) and *PpProDH1* (*proline dehydrogenase 1*) [18]. In kiwifruit (*Actinidia* spp.), AcNAC1 activates *AcALMT1* (*aluminum-activated malate transporter 1*) expression by direct binding to its promoter, and leads to a decline in citrate content [15]. In citrus, CitZAT5 (zinc finger of *arabidopsis thaliana* 5) modifies sugar accumulation and hexose proportion by positively regulating *CitSUS5* (*sucrose synthase*) and *CitSWEET6* (*sugars will eventually be exported transporter 6*) [19], and CitNAC62 has been shown to promote citric acid degradation in citrus fruits [20]. These findings strongly suggest that the TFs have an essential role in regulating sugar accumulation and acid metabolism during fruit ripening in various fruit crops.

Citrus fruit, particularly the ‘Kiyomi’ (*Citrus unshiu* × *C. sinensis*) variety, is a widely cultivated and economically significant non-climacteric fruit. Understanding the physiological and molecular mechanisms underlying its development and ripening is crucial for improving fruit quality, extending shelf life, and enhancing breeding practices. Sugars and organic acids are central determinants of fruit flavor. Integrative multi-omics analysis of their metabolic dynamics and regulatory networks during fruit development enables the identification of key transcription factors and target genes involved in sugar and acid accumulation in apple, kiwifruit and other fruit crops [21,22]. This study aims to explore the developmental regulation of sugar and organic acid metabolism in citrus fruits, focusing on the differential expression patterns of key genes involved in these processes. We integrated temporal metabolomics and transcriptomics analysis to explore the metabolic landscape of citrus fruit. The metabolomes and transcriptomes were analyzed at six representative time points (60, 90, 120, 150, 180, and 210 days after flowering, DAF) covering the cell expansion (60–90 DAF) and ripening stages (90–180 DAF) of citrus fruit development. After that, the integrative analyses of metabolome and transcriptome data provided comprehensive information on the dynamics of major metabolites and the underlying regulatory pathways, which would further guide the breeding and regulation of shelf life in citrus.

## 2. Results

### 2.1. Dynamic Change in Organic Acids and Sugars During Citrus Fruit Ripening

The size of ‘kiyomi’ fruits increased progressively during fruit development (Figure 1a). Moreover, fruit color began to turn orange at 180 DAF and became completely orange at 210 DAF (Figure 1a). In order to investigate the basis for the sweetening of the ripening flavor of ‘kiyomi’ fruits and to determine which components are mainly contained in ‘kiyomi’ fruits, we measured the content of soluble sugars and organic acids during fruit development. Citric acid was the predominant organic acid, with its content significantly higher than that of malic acid and quinic acid (Figure 1b). The content of citric acid and malic acid were very low at the early stage of fruit development, reaching a peak at 90 DAF, and then sharply declined from 90 to 180 DAF (Figure 1b). Whereas quinic acid showed a continuous decrease during fruit development (Figure 1b). Results showed that ‘kiyomi’ fruits had a high sucrose content compared to glucose and fructose contents (Figure 1b). The content of these three sugar components generally peaked at 180 DAF and then decreased slightly at 210 DAF (Figure 1b).

### 2.2. Changes in Metabolites with Fruit Development and Ripening

To generate a comprehensive insight into the metabolome dynamics and regulatory networks during ‘kiyomi’ fruit development, parallel metabolomic profiling and transcriptomic analysis were conducted. In ‘kiyomi’ fruit, a total of 1679 distinct annotated metabolites were identified, including 34.01% Flavonoids, 8.81% Phenolic acids, 8.58% Alkaloids, 8.34% Lipids, 8.22% Lignans and Coumarins, 6.31% Amino acids and derivatives, 5.96% Terpenoids, 3.28% Organic acids, 2.44% Nucleotides and derivatives, 0.42% Tannins, 0.66% Quinones, 0.06% Steroids, and 12.86% compounds of other classes (Appendix A). Pearson correlation analysis based on the profiles of 1679 metabolites demonstrated high reproducibility among biological replicates at each developmental stage, with correlation coefficients generally exceeding 0.95 (Appendix A). In addition, we performed principal component analysis (PCA) and established that the 1679 metabolites could be further divided into six groups (Figure 2a). Specifically, metabolite compounds identified in the early stage (60 DAF) were distinct from those observed in other stages, and compounds of metabolites identified from 120 and 150 DAF, 180 and 210 DAF were similar (Figure 2a). Metabolite compounds identified in the late stage (180 to 210 DAF) were distinct from those found in other stages (Figure 2a). Further compositional analysis showed that flavonoids, alkaloids, lignans and coumarins were predominant at 60 DAF, reflecting early-stage secondary metabolite accumulation (Figure 2b). In contrast, terpenoids, organic acids, and lipids were enriched at 180 and 210 DAF, suggesting their role in fruit flavor formation during ripening.

To further investigate the temporal dynamics of metabolite accumulation, orthogonal partial least-squares discriminant analysis (OPLS-DA) was conducted, identifying 1048 differentially accumulated metabolites (DAMs) across developmental stages (Appendix A). Further, k-means clustering analysis exhibited 8 distinct clusters (C1–C8) corresponding to six different developmental stages: 60 DAF (C2–C4), 90 DAF (C1), 120 DAF (C6), 150 DAF (C8), 180 DAF (C5), and 210 DAF (C7) (Figure 2c; Appendix A), suggesting that accumulation patterns of identified metabolites were diverse throughout fruit development and ripening. While inspecting the functions of clustered metabolites, we found that the clustered metabolites were involved in diverse metabolism and biosynthesis pathways (Figure 2d). Functional enrichment analysis revealed that DAMs within specific clusters were associated with distinct biosynthetic pathways. Metabolites involved in aurone, piperidine alkaloid, lignan, quinoline alkaloid, flavanol, triterpene, and saponin metabolism showed reduced abundance from 60 to 90 DAF, whereas the biosynthesis of quinones, pyridine alkaloids, vitamins, sesquiterpenoids, isoquinoline alkaloids, and monoterpenoids was enhanced from 180 to 210 DAF (Figure 2d). These findings indicate a developmental transition in secondary metabolic programs underlying flavor changes in ‘kiyomi’ fruit ripening.

### 2.3. Hormone Dynamics and Their Regulatory Associations with Metabolites During Citrus Fruit Development

As a typical non-climacteric fruit, ‘kiyomi’ exhibited dynamic changes in hormone levels during development. Abscisic acid (ABA) and its conjugate, ABA-glucose ester (ABA-GE), initially decreased and then increased from 120 to 210 days after anthesis, reaching peak levels at 210 DAF (Figure 3a). Notably, ABA levels were also relatively high during the early stages of fruit development, suggesting that ABA may play biological roles beyond its well-known function in promoting ripening. The content of 1-aminocyclopropane-1-carboxylic acid (ACC), a precursor of ethylene, increased sharply between 60 and 120 DAF, followed by a gradual decline from 150 to 210 DAF (Figure 3a). These temporal hormone patterns imply that both ABA and ACC may be involved in regulating the ripening process in ‘kiyomi’ citrus. To explore potential regulatory relationships between hormones and metabolic changes, we analyzed metabolite–hormone correlations. Metabolites exhibiting strong positive or negative correlations with ABA (|PCC| ≥ 0.8) were identified, suggesting that ABA is involved in both secondary metabolism and nitrogen/carbon reallocation during development (Figure 3b; Appendix A). Similarly, metabolites significantly correlated with ACC (|PCC| ≥ 0.8) were associated with flavor-related metabolism and ripening-associated reprogramming (Figure 3c; Appendix A). Taken together, these results indicate that ‘kiyomi’ fruit ripening primarily occurs between 180 and 210 DAF, a period characterized by marked changes in terpenoids, alkaloids, flavonoids, and organic acids. Among the regulatory signals, ABA appears to play a central role in coordinating metabolic transitions during fruit maturation.

### 2.4. Transcriptomic Changes During Fruit Development and Ripening

To investigate transcriptomic dynamics during the development of citrus fruit, the transcriptomes at the six development stages were investigated. Pearson correlation analysis among RNA-seq samples showed high reproducibility of biological replicates at each developmental stage, supporting the reliability of transcriptomic data (Appendix A). Principal component analysis (PCA) showed clear separation of transcriptome profiles across the six developmental stages, indicating significant differences during fruit ripening (Figure 4a). The genes of which the average Transcripts Per Million (TPM) > 0 were defined as expressed and genes with the absolute value of Log2 (Fold Change of TPM) > 0.585 were defined as differentially expressed [23,24]. Then we made a Z-score normalized expression heatmap clustered by genes. There were the most differentially expressed genes in 180 DAF vs. 210 DAF group, suggesting a dramatic change at the transcriptome level during maturation. In contrast, transcriptome changes were relatively small in the 120 DAF vs. 150 DAF (Figure 4b and Appendix A. K-means clustering of gene expression patterns identified six gene groups (Figure 4c; Appendix A). The largest cluster (C3) was enriched for genes involved in epigenetic regulation and cytokinesis via cell plate formation. In contrast, clusters positively correlated with fruit ripening (C1 and C2) were enriched for genes associated with organic acid catabolism, vacuolar transport, and terpenoid metabolism. These findings are consistent with observed biochemical changes during ripening, such as decreased organic acid content and increased carbohydrate accumulation. Additionally, 584 DEGs (5.80%) were annotated as transcription factors (TFs) (Figure 4d; Appendix A). Among these, members of the ERF, bHLH, and MYB families were predominant, suggesting their potential roles in regulating fruit ripening processes.

### 2.5. Transcriptional Regulatory Networks Underlying Soluble Sugar and Organic Acid Metabolism During Citrus Fruit Ripening

To better understand the transcriptional regulation of metabolic reprogramming during citrus fruit development and ripening, weighted gene co-expression network analysis (WGCNA) was performed based on differentially expressed genes (DEGs). Eleven co-expression modules were identified according to expression similarity (Figure 5a; Appendix A), with the majority of DEGs clustered in the turquoise module. Module-trait correlation analysis (Figure 5b) showed that the turquoise module was positively correlated with citric acid content, whereas the blue and black modules were negatively correlated with citric acid but positively associated with glucose, fructose, and sucrose, which are key flavor components during citrus fruit ripening.

Soluble sugars are among the most important components contributing to the characteristic flavor of citrus fruit. Interestingly, and the accumulation of sucrose, fructose, glucose was highly correlated with blue and black modules (Figure 5b). To generate the regulatory network associated with soluble sugars metabolism, we examined the structural genes involved in soluble sugar metabolic pathway identified in blue and black modules (Figure 5b). To construct a regulatory network associated with soluble sugar metabolism, we examined structural genes within these modules (Figure 6a). Fifteen sugar-related genes were identified, including three *invertases* (*INV*), two *sucrose synthases* (*SUS*), three *sucrose transporters* (*SUT*), one *SWEET transporter*, one *sucrose phosphate phosphatase* (*SPP*), two *sucrose phosphate synthetase* (*SPS*) genes, one *tonoplast monosaccharide transporter* (*TMT*) and two *fructose-1,6-bisphosphates* (*FBP*), all of which were structural genes involved in sugar synthesis and transport and highly correlated with sugar accumulation. By integrating transcript abundance profiles with predicted promoter-binding affinities, we identified 79 transcription factors, mainly belonging to the ERF, NAC, MYB, and HSF families, that were strongly associated with the expression of these structural genes (Figure 6a; Appendix A). These transcription factors likely serve as putative regulators of soluble sugar metabolism during fruit ripening.

Organic acids such as citric acid, malic acid, and quinic acid also contribute substantially to citrus flavor. Citric acid is the predominant organic acid in many citrus species, including the ‘kiyomi’ cultivar. Several structural genes, including *phosphoenolpyruvate carboxylase* (*PEPC*), *fructose-1,6-diphosphatase* (*FBP*), *aluminum-activated malate transporters* (*ALMT*), *citrate synthase* (*CS*), *ATP-citrate lyase* (*ACL*), *aconitase* (*ACO*), *malate dehydrogenase* (*MDH*), *isocitrate dehydrogenase* (*IDH*), *glutamate decarboxylase* (*GAD*), *dicarboxylate transporter* (*DiT*), and *vacuolar-ATPases* (*PH/VHA/VHP*), are thought to be involved in the biosynthesis and degradation of malic acid and citric acid (Figure 6b). To identify candidate regulators of citric acid metabolism, we focused on the blue and black modules, which were negatively correlated with citric acid content. Within these modules, five structural genes, *ACO3*, *NAD-IDH3*, *NADP-IDH3*, *GAD1*, and *GABA-T*, were identified as being strongly negatively correlated with citric acid levels throughout fruit ripening (Figure 6b). Further analysis of the turquoise module identified 84 transcription factors, mainly from the MYB, bZIP, NAC, and HSF families, whose expression patterns were associated with these citric acid-related genes. A putative transcriptional regulatory network was constructed based on expression correlation (Figure 6b; Appendix A).

## 3. Discussion

Citrus fruits are highly valued for their nutritional qualities and health benefits, making them a key focus of horticultural research. Fruit ripening involves tightly regulated metabolic shifts, particularly in sugar and organic acid accumulation, together with aroma-related volatiles, which contribute significantly to flavor and overall fruit quality [25]. In citrus, sucrose, glucose, and fructose are the predominant soluble sugars, and their metabolism is governed by key enzymes including sucrose phosphate synthase (SPS), sucrose synthase (SUS), hexokinase (HXK), fructokinase (FRK), and invertases (CINV) [26]. Likewise, organic acid metabolism, particularly citric acid, is mediated by citrate synthase (CS), aconitase (ACO), and isocitrate dehydrogenase (IDH) [27]. Despite these advances, the dynamics of soluble sugars, organic acids, and related genes during fruit development remain unclear in many citrus varieties. In this study, we integrated metabolomic, hormonal, and transcriptomic analyses to dissect the developmental regulation of flavor-related metabolic pathways in ‘Kiyomi’ citrus.

Citrus fruit ripening is accompanied by the synthesis and degradation of primary and secondary metabolites. In this study, we identified 1679 distinct metabolites in ‘kiyomi’ fruit, with significant changes observed across different developmental stages (Figure 2b). The early stage (60 DAF) was characterized by high levels of flavonoids, alkaloids, and lignans, while the ripening stages (180 and 210 DAF) were marked by an increase in terpenoids, organic acids, and lipids (Figure 2b). These findings are consistent with previous studies on citrus fruit, which have shown that secondary metabolites such as flavonoids and terpenoids play important roles in fruit development and flavor formation [28,29]. The accumulation of specific metabolites, such as citric acid and sucrose, is particularly important for the characteristic flavor and nutritional value of fruit [22]. Sucrose metabolism is a key process in fruit ripening. Our study identified several differentially expressed genes (DEGs) involved in sucrose metabolism, including *SPS* and *SPP*. Up-regulation of *SPP* and *SPS* genes suggests that products of sugar metabolism flow toward sucrose synthesis and storage. Citric acid metabolism is tightly linked to the tricarboxylic acid (TCA) cycle. The balance between citric acid synthesis and degradation is crucial for regulating citric acid accumulation in fruit. During ripening stage, citric acid is released from vacuoles and metabolized into isocitrate by *ACO* and into 2-oxoglutarate by *NADP IDH* [27,30,31]. In our study, citric acid peaked at 90 DAF while decreasing sharply from 150 to 210 DAF and was accompanied by the up-regulation of *ACO*, *IDH* and *GAD* genes, suggesting that the degradation of citric acid via the GABA pathway is a key process during fruit ripening.

Abscisic acid (ABA) and ethylene are key hormones involved in fruit ripening. In this study, ABA levels increased progressively from 120 to 210 DAF, while ACC, an ethylene precursor, peaked early (60–120 DAF) and declined thereafter. These results underscore the contrasting temporal roles of these hormones, with ABA likely serving as the primary regulator of ripening in this non-climacteric fruit, consistent with findings in strawberry, grape, and citrus [5,32,33]. Collectively, our data support a hierarchical model in which ABA operates as a master switch that gates the transcriptional reprogramming of soluble sugar and organic acid metabolism. Elevated ABA at 120–210 DAF directly or indirectly activates NAC/ERF-type TFs. These TFs, in turn, trans-activate downstream effector genes such as *ACO*, *IDH*, *GAD1*, *GABA-T*, *SPS*, and *SWEET*, thereby accelerating citrate catabolism via the GABA shunt and promoting soluble sugar through sugar synthesis and transport. Our correlation analysis revealed that ABA is closely associated with the regulation of secondary metabolism and the regulation of soluble sugar and organic acid accumulation, which are essential for fruit ripening. These findings highlight the complex interplay between ABA in citrus fruit ripening and suggest that ABA may have a more direct role in modulating ripening, similar to the findings in mango fruit [34].

The transcriptomic analysis revealed significant changes in gene expression across the developmental stages of ‘kiyomi’ fruit. The highest number of differentially expressed genes was observed between 180 DAF and 210 DAF, indicating that gene expression changes dramatically during the ripening stage. Functional enrichment analysis showed that these DEGs were involved in pathways such as organic acid catabolism, vacuolar transport, and terpenoid metabolism, aligning with the observed biochemical shifts during ripening.

Weighted gene co-expression network analysis (WGCNA) further revealed the complex regulatory networks associated with soluble sugar and organic acid metabolism. The correlation between gene expression and metabolite accumulation highlights the importance of specific TFs in modulating the biosynthesis of key metabolites. Among the TFs, members of the ERF and NAC families were enriched in the DEGs during fruit ripening processes. Previous studies have reported CitERF6 contributed to citric acid degradation via upregulation of *CitAclα1* in citrus [35]. ERF could also promote sugar accumulation via trans-activation of *SWEET* gene expression in both citrus and grape [36,37]. Members of the NAC TF family are known regulators of fruit ripening and senescence [38,39,40], as exemplified by CitNAC62, which cooperates with CitWRKY1 to drive citric acid degradation via *CitACO3* [20], MdNAC5 in apple, which regulates fructose accumulation through *MdTST2* and *MdNINV6* [41], and PpNAP4 in peach, which modulates sucrose accumulation by activating *PpSUS1* and *PpSPS2* [42]. In conclusion, the identification of ERF and NAC transcription factors in the blue and black co-expression modules suggests their potential roles in regulating soluble sugar and organic acid metabolism during citrus fruit ripening.

The findings from this study have important implications for citrus breeding and postharvest management. Understanding the key metabolites and genes involved in ripening can help in the development of new cultivars with improved flavor, nutritional value, and shelf life. For example, the identification of genes associated with sugar and organic acid metabolism can be targeted for genetic improvement. Moreover, the insights into hormonal regulation can inform postharvest treatments to enhance fruit quality and delay senescence. Exogenous application of ABA or its inhibitors may be explored to modulate ripening progression and extend shelf life.

## 4. Materials and Methods

### 4.1. Plant Materials

Six-year-old ‘kiyomi’ (*Citrus unshiu* × *C. sinensis*) trees from an orchard (Hubei Academy of Agricultural Sciences, Wuhan, Hubei Province, China; 30.484075° N, 114.320471° E) were used in the study. Fruits were randomly sampled at six developmental stages: 60, 90, 120, 150, 180, and 210 days after flowering (DAF), corresponding to the cell expansion stage (60–90 DAF) and the fruit ripening stage (90–180 DAF). Juice sacs (pulp) were collected for subsequent analyses. For each stage, three biological replicates were collected, each comprising ten fruits. Samples were immediately frozen in liquid nitrogen and stored at −80 °C until further use.

### 4.2. Quantification of Abscisic Acid (ABA) and 1-Aminocyclopropane-1-Carboxylic Acid (ACC)

Extraction and quantification of ABA and ACC were performed by Wuhan Metware Biotechnology Co., Ltd. (Wuhan, China) using the AB Sciex QTRAP 6500 LC-MS/MS platform (Shimadzu, Kyoto, Japan; Applied Biosystems, Foster City, CA, USA). Briefly, approximately 500 mg of receptacle tissue was ground and homogenized in 1 mL of methanol/water/formic acid (15:4:1, *v*/*v*/*v*), followed by centrifugation at 12,000 rpm for 5 min at 4 °C. The resulting supernatant was evaporated to dryness, reconstituted in 100 μL of 80% methanol (*v*/*v*), and filtered through a 0.22 μm membrane prior to LC-MS/MS analysis [43,44]. Three independent biological replicates were analyzed.

### 4.3. Widely Targeted Metabolomics Analysis

Extraction and metabolite profiling were performed by Wuhan Metware Biotechnology Co., Ltd. (Wuhan, China) following standard protocols [45]. Briefly, approximately 100 mg of fruit pulp was extracted using 70% aqueous methanol, and the extracts were analyzed using a UPLC-ESI-MS/MS system (Shimadzu, Kyoto, Japan; Applied Biosystems, Foster City, CA, USA). Metabolite separation was followed by electrospray ionization tandem mass spectrometry (ESI-MS/MS, Applied Biosystems, Foster City, CA, USA). Three biological replicates were analyzed for each developmental stage.

### 4.4. Sucrose and Organic Acid Determination

The contents of sucrose and citric acid in 1 g of citrus juice sacs were determined using a gas chromatograph (7890B; Agilent Technologies, Santa Clara, CA, USA), following the method described by Wang et al. [46]. Three independent biological extractions were performed for each sample.

### 4.5. RNA Extraction and Transcriptomic Analysis

Total RNA was extracted from fruit pulp samples collected at six developmental stages (60, 90, 120, 150, 180, and 210 days after flowering, DAF) of ‘Kiyomi’, with three biological replicates per stage. RNA extraction was performed using the HiPure HP Plant RNA Mini Kit (Magen, Guangzhou, China) following the manufacturer’s instructions. RNA sequencing was performed using the Illumina NovaSeq 6000 platform by Personal Biotechnology Co., Ltd. (Shanghai, China), and transcriptomic data were obtained from three independent biological replicates. Raw reads were processed using fastp (v0.23.1) using a sliding window approach with a phred score quality threshold of 33 (–cut-right -q 33) with the --detect_adapter_for_pe parameter to remove adapter sequences, trim low-quality bases, and filter out poor-quality reads [47]. Clean reads were then aligned to the *Citrus sinensis* reference genome (v3.0, downloaded from http://citrus.hzau.edu.cn/ (accessed on 12 December 2023)) using STAR (v2.4.2a) [48]. Gene-level read counts were quantified using FeatureCounts (v2.0.1), based on the genome annotation of *C. sinensis* v3.0 [49]. Differentially expressed genes (DEGs) were identified using the DESeq2 package (v1.28.1) in R (v4.3.2), with thresholds of adjusted *p*-value (Padj) < 0.05 and |log_2_FoldChange| > 0.585 [50]. Transcripts per kilobase million (TPM) values, derived from the count matrix, were used for expression level estimation, principal component analysis (PCA), and hierarchical clustering. PCA plots were then plotted with ggplot2 package in R (v4.3.2). K-means clustering was performed using Z-scaled TPM values in R (v4.3.2), and gene expression heatmaps were generated using the ComplexHeatmap package (v2.4.3) [51]. Gene Ontology (GO) enrichment analysis was conducted using AgriGO v2.0 (http://systemsbiology.cau.edu.cn/agriGOv2/ (accessed on 2 March 2024)). Transcription factor (TF) identification and classification were based on PlantTFDB (http://planttfdb.gao-lab.org/ (accessed on 8 September 2024)) [52]. Enrichment analysis of TF families among DEGs was performed using the enricher function in the clusterProfiler package (v3.16.1) [53].

### 4.6. Weighted Correlation Network Analysis and Gene Network Visualization

To construct gene co-expression modules, normalized gene expression data (TPM) were first filtered to remove lowly expressed and invariant genes. Specifically, genes were retained if their expression values exceeded 1 in more than 90% of the samples. Subsequently, genes with high variability were selected using the median absolute deviation (MAD) method, resulting in a total of 13,517 genes for downstream analysis.

The co-expression modules were obtained using automatic network construction function (blockwiseModules) with default parameters. The position frequency matrices (PFMs) of transcription factors were obtained from plantTFDB [54], which were used to predict cis-acting element information in the promoter region of the structural genes (1000 bp upstream and 200 bp downstream of the transcription start site) under the condition of *p* value ≤ 1 × 10^−4^ by FIMO [55]. The transcriptional regulatory networks were generated by combining the Pearson correlation coefficient (PCC > 0.8) between structural genes and transcription factors and the availability of cis-element binding sites present in the promoter regions of structural genes in the same module. The networks were visualized by CYTOSCAPE (v.3.7.2) [56].

## 5. Conclusions

In conclusion, this study provides a comprehensive analysis of the metabolite dynamics, hormonal regulation, and transcriptomic changes throughout the ripening of ‘Kiyomi’ citrus fruit. The results suggest the potential involvement of ABA and specific transcription factors in mediating ripening-associated metabolic transitions and offer promising targets for genetic improvement and postharvest strategies.

## Figures and Tables

**Figure 1 plants-14-02751-f001:**
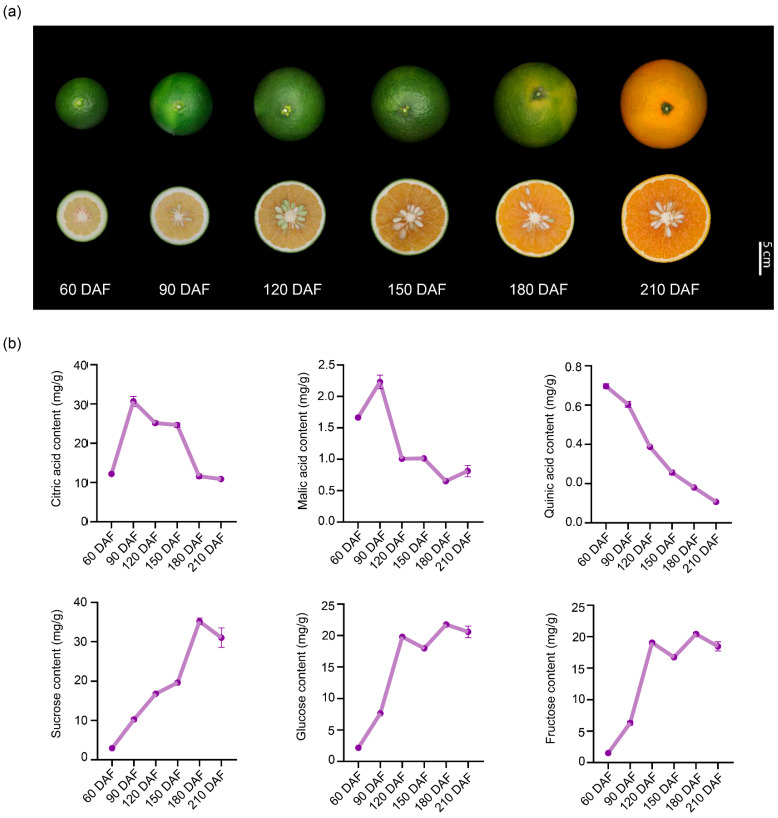
Morphological characteristics and dynamic changes in major organic acids and soluble sugars during ‘kiyomi’ fruit development. (**a**) Changes in ‘kiyomi’ fruit at six developmental stages: 60, 90, 120, 150, 180, and 210 days after flowering (DAF). Scale bar = 5 cm. (**b**) Citric acid, malic acid, quinic acid, sucrose, glucose, and fructose levels during fruit development. Data were expressed as means ± SD of three replicates.

**Figure 2 plants-14-02751-f002:**
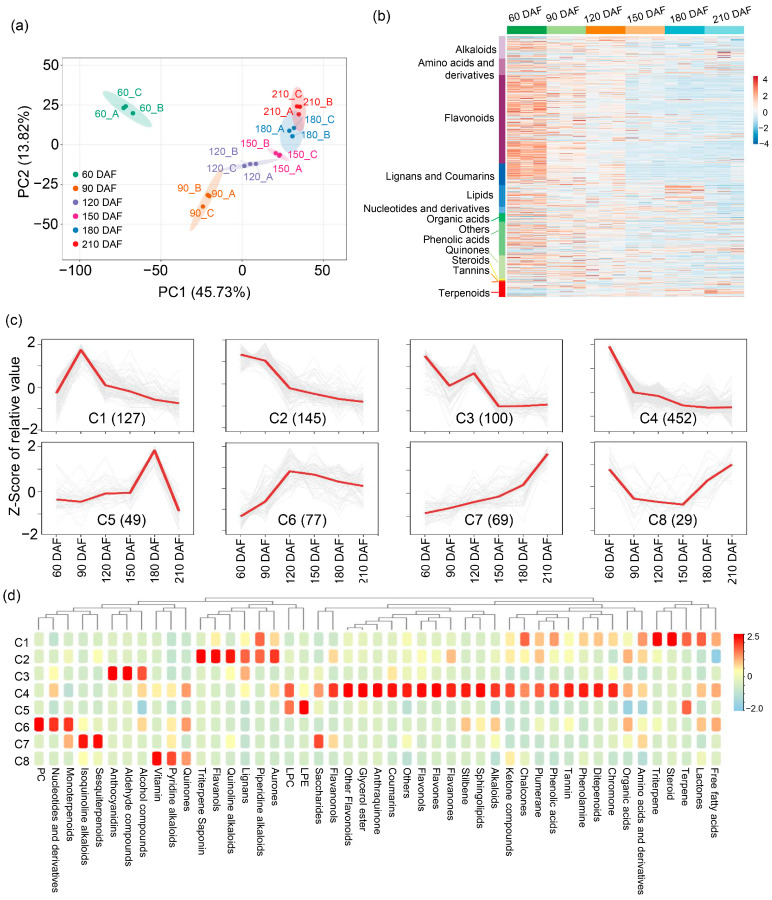
Metabolomic profiling of ‘kiyomi’ fruit at six developmental stages. (**a**) Principal component analysis (PCA) based on the metabolite profiles of samples collected at 60, 90, 120, 150, 180, and 210 DAF. (**b**) Heat map of 1679 identified metabolites across developmental stages. (**c**) K-means clustering of 1048 differentially accumulated metabolites (DAMs). The x-axis represents developmental stages; the y-axis shows Z-score-normalized relative abundances. The number in parentheses indicates the number of DAMs in each cluster (C1–C8). (**d**) Metabolite enrichment analysis of the eight clusters.

**Figure 3 plants-14-02751-f003:**
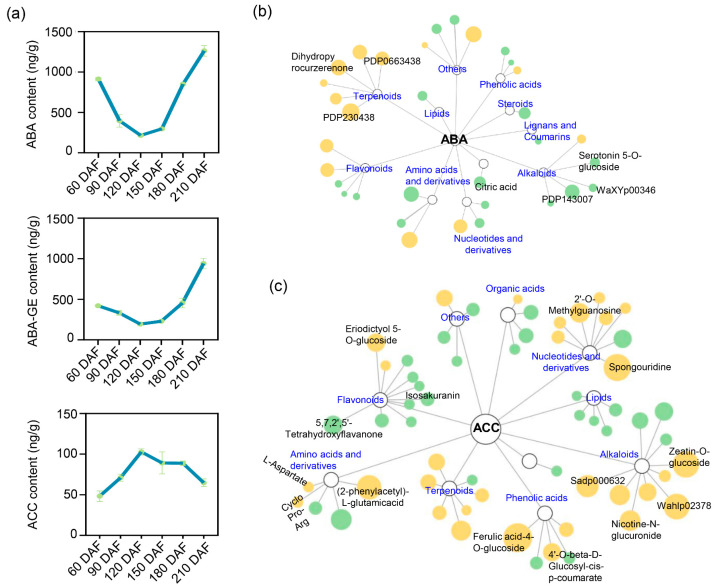
Hormone dynamics and their regulatory associations with metabolites during ‘kiyomi’ fruit development. (**a**) Temporal changes in the levels of abscisic acid (ABA), ABA-glucose ester (ABA-GE), and 1-aminocyclopropane-1-carboxylic acid (ACC) across six developmental stages. (**b**,**c**) Correlation networks between ABA (**b**) or ACC (**c**) and associated metabolites. orange nodes indicate metabolites positively correlated with hormone levels, while green nodes indicate negative correlations. Node size reflects the absolute value of the Pearson correlation coefficient (|PCC| ≥ 0.8), indicating the strength of association.

**Figure 4 plants-14-02751-f004:**
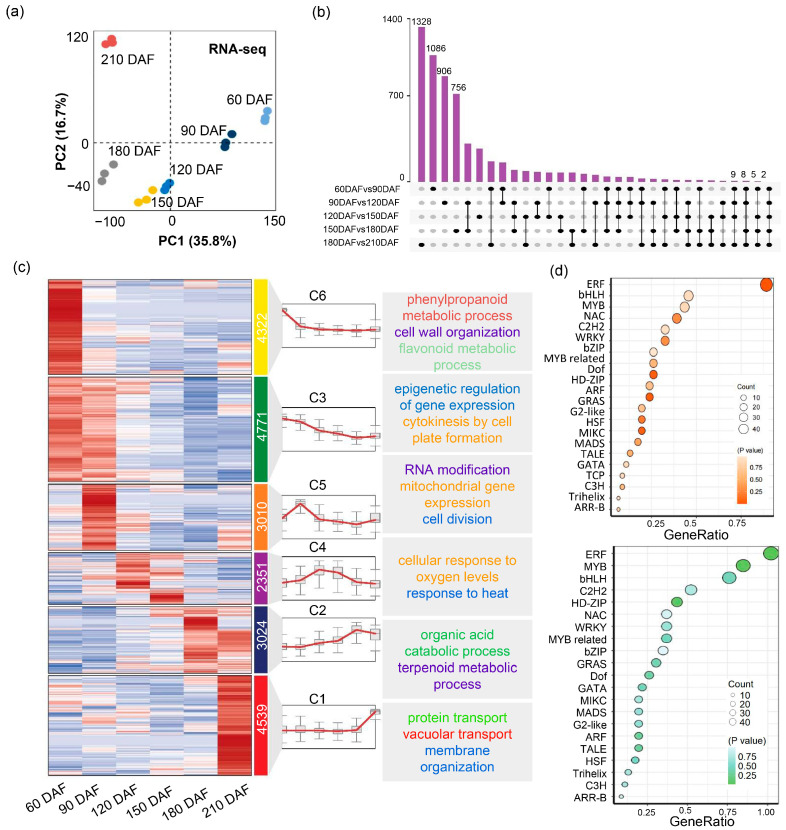
Global expression profile of six developmental stages in ‘kiyomi’ fruit. (**a**) Principal component analysis (PCA) of transcriptome data across six developmental stages of ‘kiyomi’ fruit. (**b**) Upset plot of upregulated DEGs in pairwise comparisons between consecutive fruit developmental stages. (**c**) K-means clustering of DEGs into six major expression clusters (C1–C6), along with representative expression trends and associated GO terms for each cluster. (**d**) Transcription factor (TF) family enrichment analysis based on DEGs. The upper panel shows enriched TF families among upregulated DEGs; the lower panel shows those among downregulated DEGs.

**Figure 5 plants-14-02751-f005:**
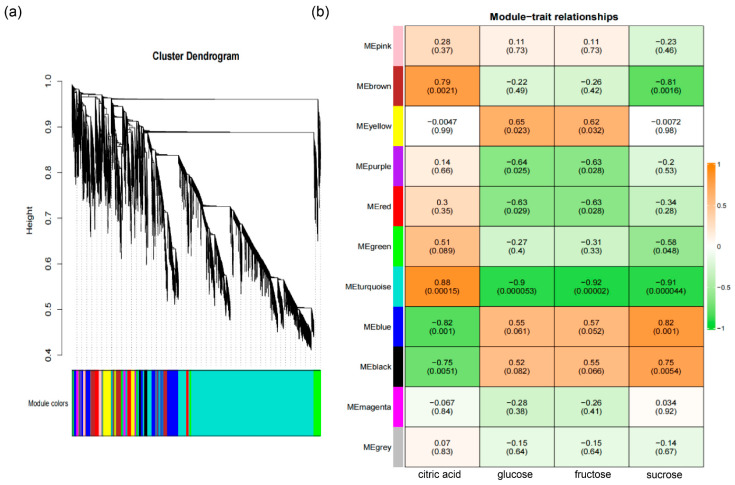
WGCNA analysis of gene co-expression modules and their associations with major metabolites during ‘kiyomi’ fruit development. (**a**) Dendrogram of gene co-expression modules identified by weighted gene co-expression network analysis (WGCNA). Eleven distinct modules were detected, each represented by a unique color. (**b**) Heatmap of correlations between gene modules and key metabolites (citric acid, glucose, fructose and sucrose). Each cell indicates the Pearson correlation coefficient and the corresponding *p*-value (in parentheses). Orange and green indicate positive and negative correlations, respectively.

**Figure 6 plants-14-02751-f006:**
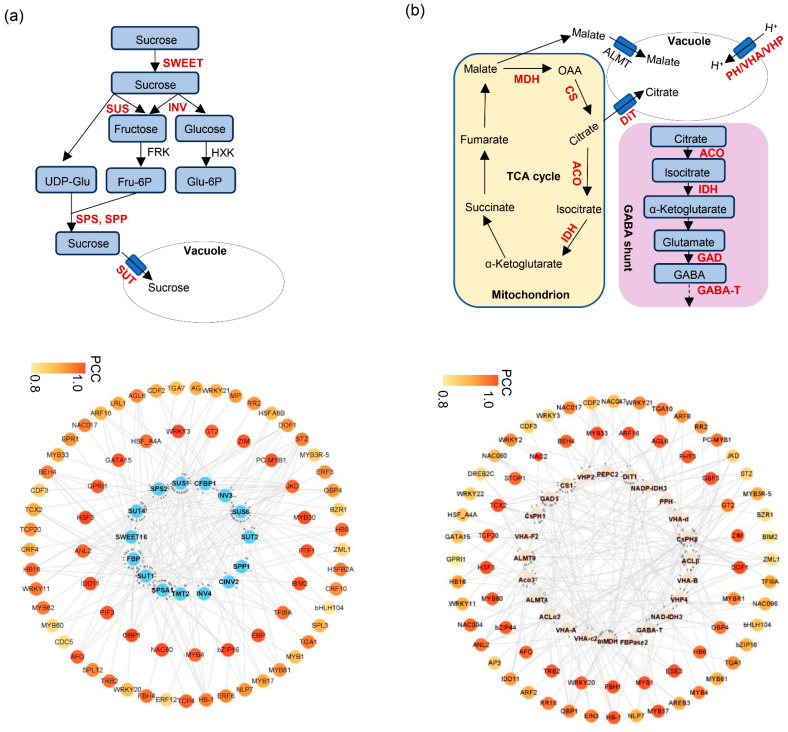
Transcriptional regulatory networks associated with soluble sugar and citric acid metabolism during ‘kiyomi’ fruit development. (**a**) Upper: schematic diagram of the soluble sugar metabolic pathway. *SWEET*, *sugars will eventually be exported transporter*; *INV*, *invertase*; *SUS*, *sucrose synthase*; *SUT*, *sucrose transporter*; *SPS*, *sucrose phosphate synthetase*. (**a**) Lower: blue circles represent structural genes involved in soluble sugars metabolism during fruit development and ripening. (**b**) Upper: schematic diagram of the citric acid biosynthesis and degradation pathway. *CS*, *citrate synthase*; *ACO*, *aconitase*; *MDH*, *malate dehydrogenase*; *IDH*, *isocitrate dehydrogenase*; *GAD*, *glutamate decarboxylase*; *DiT*, *di-carboxylate transporter*; *PH/VHA/VHP*, *vacuolar-ATPases*. (**b**) Lower: pink circles represent structural genes involved in citric acid biosynthesis and degradation.

## Data Availability

All of the data are contained within the article.

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
