# Peer review of "Metabolomic and Transcriptomic Analyses Provide Insights into Metabolic Networks During Kiyomi Tangors Development and Ripening"

_plants, 2025, doi:10.3390/plants14172751_

Round 1
Reviewer 1 Report
Comments and Suggestions for Authors
This is a comprehensive and well conducted study evaluating the changes in the metabolome and transcriptome during six stages of 'Kiyomi' tangors fruit development, with special emphasis on the changes related to sugar and acid metabolisms and their putative transcriptional regulation.
I just have a few minor comments listed below:
1. Title – change the words "Citrus Fruit" to "'Kiyomi' tangors".
2. In the Introduction (line 36) and Discussion (line 277) chapters – it is necessary to indicate that during ripening there are changes in taste and aroma and not only in the taste parameter.
3. Results, line 192 – please spell out what does TPM stand for?
4. Results, Fig. 4d – the same groups of ERF, bHLH, MYB, NAC, C2H2 and WRKY TF's are most abundant in both the up-regulated and down-regulated regulons. Please attribute to this issue.
5. Discussion, line 339 – the authors indicate that changes in ERF and NAC TF's are most important. Please elaborate on this issue.
6. M&M, section 4.5, line 379 – please indicate from which tissue was the RNA extracted?
7. Discussion – I think it is too ambitious to indicate that they identified the "central roles of ABA and specific transcription factors in mediating ripening-associated metabolic transitions". This assumption yet needs to be proved by functional genomic approaches. The author just suggest the possible involvement of ABA and these TFs.
Author Response
Dear Reviewer,
Thanks for considering our manuscript and giving us helpful comments. We have made the necessary revisions to the manuscript and have provided detailed responses below to clarify the points you raised. We believe these changes significantly improve the clarity and quality of our work.
Fang Song
2025.8.27
Point-by-point response to Comments and Suggestions for Authors
Comments 1: Title – change the words "Citrus Fruit" to "'Kiyomi' tangors".
Response 1: Thank you for your suggestion. We have revised the title and changed the words “Citrus Fruit” to “‘Kiyomi’ tangors”.
Comments 2: In the Introduction (line 36) and Discussion (line 277) chapters – it is necessary to indicate that during ripening there are changes in taste and aroma and not only in the taste parameter.
Response 2: Thank you very much for this helpful suggestion. We agree with the reviewer that fruit ripening involves not only changes in taste but also in aroma. Accordingly, we have revised the sentences in lines 35-36 (Introduction): “Fruit ripening is a highly complex and well-regulated biological process, involving significant changes in taste (sweetness and acidity), aroma, texture (softening and firmness), and appearance (color)” and lines 290-291 (Discussion): “Fruit ripening involves tightly regulated metabolic shifts, particularly in sugar and organic acid accumulation, together with aroma-related volatiles, which contribute significantly to flavor and overall fruit quality.”
Comments 3: Results, line 192 – please spell out what does TPM stand for?
Response 3: Thank you for this suggestion. We have revised the text to spell out the abbreviation TPM at its first occurrence (lines 199-201): “The genes of which the average Transcripts Per Million (TPM) > 0 were defined as expressed, and genes with the absolute value of Log2 (Fold Change of TPM) > 0.585 were defined as differentially expressed.”
Comments 4: Results, Fig. 4d – the same groups of ERF, bHLH, MYB, NAC, C2H2 and WRKY TF's are most abundant in both the up-regulated and down-regulated regulons. Please attribute to this issue.
Response 4: Thank you for this comment. These transcription factor families are large and contain diverse members with distinct, sometimes even opposite, regulatory functions. Therefore, it is reasonable that ERF, bHLH, MYB, NAC, C2H2, and WRKY appear as the most abundant groups in both the up-regulated and down-regulated regulons, reflecting their complex and stage-specific roles in fruit ripening. Similar results have also been reported in Citrus reticulata ‘Chachi’ flavonoid biosynthesis studies.
Zhu C, You C, Wu P, et al. The gap-free genome and multi-omics analysis of Citrus reticulata ‘Chachi’reveal the dynamics of fruit flavonoid biosynthesis. Horticulture Research, 2024, 11(8): uhae177.
Comments 5: Discussion, line 339 the authors indicate that changes in ERF and NAC TF's are most important. Please elaborate on this issue.
Response 5: Thank you for this helpful suggestion. More information about ERF and NAC TFs were added in the discussion section in Line 348-351. “The correlation between gene expression and metabolite accumulation highlights the importance of specific TFs in modulating the biosynthesis of key metabolites. Among the TFs, members of the ERF and NAC families were enriched in the DEGs during fruit ripening processes.”
Comments 6: M&M, section 4.5, line 379 – please indicate from which tissue was the RNA extracted?
Response 6: Thank you for pointing this out. We have revised the text in lines 403-405 to clarify the tissue used for RNA extraction: “Total RNA was extracted from fruit pulp samples collected at six developmental stages (60, 90, 120, 150, 180, and 210 days after flowering, DAF) of ‘Kiyomi’, with three biological replicates per stage.”
Comments 7: Discussion – I think it is too ambitious to indicate that they identified the "central roles of ABA and specific transcription factors in mediating ripening-associated metabolic transitions". This assumption yet needs to be proved by functional genomic approaches. The author just suggest the possible involvement of ABA and these TFs.
Response 7: Many thanks for the nice suggestion. Therefore, We have revised the text in lines 446-448: “The results suggest the potential involvement of ABA and specific transcription factors in mediating ripening-associated metabolic transitions and offer promising targets for genetic improvement and postharvest strategies.”.
Reviewer 2 Report
Comments and Suggestions for Authors
The authors performed a integrative analysis (physiological and bioinformatics) on citrus fruits.
Some suggestions are included in the following paragraphs to improve the manuscript.
L54. Please include the word “gene”… FaNCED1 gene.. Moreover, al the names of the genes should be in italics as in line 67. Please review this in the whole manuscript.
L79-84. Before explaining the objective of the study. Please include other studies about the integration of omics data about the regulation of sugar and organic acid metabolism. Even if no studies can be found on citrus fruits. Several studies can be found in other plant species such as tomato, soursop, avocado, among others.
Results
L146. In the Figure 2b, the word Nucleotides and derivatives touch the Figure 1a. Please edit this
L269-273. Authors should include more information of the Fig 6a and Fig 6b. Authors need to explain the interaction between the genes and metabolites or citric acid levels found.
Are these correlated? What is the meaning of this correlation?
Discussion
L349. After this paragraph. I suggest that the authors explain more about the integrative analysis. It is important to mention how these gene-metabolite-hormone are related. Furthermore, explain or hypothesize how the physiological parameters measured are associated with the gene expression.
This will enhance the findings of this study.
Materials and methods
L384-385. Please include the parameters (Q<20)??
L390. Usually, the default parameter for the log2foldchange is > 1, which is that at least 2 times a gene is expressed. The authors used 0.585, why do the authors used this cut-off value?
L414. Please include the name of the softwares or the packages in R used to create the images.
Author Response
Dear Reviewer,
Thank you very much for your valuable feedback. We have made the necessary revisions to the manuscript and have provided detailed responses below to clarify the points you raised. We believe these changes significantly improve the clarity and quality of our work.
Fang Song
2025.8.27
Point-by-point response to Comments and Suggestions for Authors
Comments 1: L54. Please include the word “gene”… FaNCED1 gene.. Moreover, al the names of the genes should be in italics as in line 67. Please review this in the whole manuscript.
Response 1: Thank you for this helpful suggestion. We have revised the text at line 54 to read “FaNCED1 gene” and carefully checked the entire manuscript to ensure that all gene names are presented in italics consistently.
Comments 2: L79-84. Before explaining the objective of the study. Please include other studies about the integration of omics data about the regulation of sugar and organic acid metabolism. Even if no studies can be found on citrus fruits. Several studies can be found in other plant species such as tomato, soursop, avocado, among others.
Response 2: Many thanks for the valuable suggestion. We have added “Sugars and organic acids are central determinants of fruit flavor. Integrative multi-omics analysis of their metabolic dynamics and regulatory networks during fruit development enables the identification of key transcription factors and target genes involved in sugar and acid accumulation in apple, kiwifruit and other fruit crops” in lines 79-83 of the new manuscript. In addition, the related references are also added.
Wang, R.; Shu, P.; Zhang, C.; Zhang, J.; Chen, Y.; Zhang, Y.; Du, K.; Xie, Y.; Li, M.; Ma, T.; et al. Integrative Analyses of Metabolome and Genome-Wide Transcriptome Reveal the Regulatory Network Governing Flavor Formation in Kiwifruit (Actinidia Chinensis). New Phytol. 2022, 233, 373–389.
Comments 3: L146. In the Figure 2b, the word Nucleotides and derivatives touch the Figure 1a. Please edit this
Response 3: Thank you for your suggestion. We have revised Figure 2b and moved the label “Nucleotides and derivatives” to avoid overlapping with Figure 2a.
Comments 4: L269-273. Authors should include more information of the Fig 6a and Fig 6b. Authors need to explain the interaction between the genes and metabolites or citric acid levels found.
Are these correlated? What is the meaning of this correlation?
Response 4: Thanks for your critical judgement. We revised the result of Fig 6a and Fig 6b to explain the correlation between genes and metabolites. See Line 249-254 and Line 272-277 in the new manuscript.
“Fifteen sugar-related genes were identified, including three invertases (INV), two sucrose synthases (SUS), three sucrose transporters (SUT), one SWEET transporter, one sucrose phosphate phosphatase (SPP), two sucrose phosphate synthetase (SPS) genes, one tonoplast monosaccharide transporter (TMT) and two fructose-1, 6-bisphosphates (FBP), all of which were structural genes involved in sugar synthesis and transport and highly correlated with sugar accumulation.”
“Several structural genes, including phosphoenolpyruvate carboxylase (PEPC), fructose-1,6-diphosphatase (FBP), aluminum-activated malate transporters (ALMT), citrate synthase (CS), ATP-citrate lyase (ACL), aconitase (ACO), malate dehydrogenase (MDH), isocitrate dehydrogenase (IDH), glutamate decarboxylase (GAD), dicarboxylate transporter (DiT), and vacuolar-ATPases (PH/VHA/VHP), are thought to be involved in the biosynthesis and degradation of malic acid and citric acid.”
Comments 5: L349. After this paragraph. I suggest that the authors explain more about the integrative analysis. It is important to mention how these gene-metabolite-hormone are related. Furthermore, explain or hypothesize how the physiological parameters measured are associated with the gene expression.
This will enhance the findings of this study.
Response 5: Thank you for your suggestion. We had added a paragraph to discuss the relation among genes, metabolites and hormones in the discussion section. See Line 328-334 in the new manuscript.
Collectively, our data support a hierarchical model in which ABA operates as a master switch that gates the transcriptional reprogramming of soluble sugar and organic acid metabolism. Elevated ABA at 120-210 DAF directly or indirectly activates NAC/ERF-type TFs. These TFs, in turn, trans-activate downstream effector genes such as ACO, IDH, GAD1, GABA-T, SPS, and SWEET, thereby accelerating citrate catabolism via the GABA shunt and promoting soluble sugar through sugar synthesis and transport.
Comments 6: L384-385. Please include the parameters (Q<20)?
Response 6: Thank you for pointing this out. We actually used a more stringent filtering criterion with fastp (-q 33), which is stricter than Q < 20. We added the parameters in the Materials and method section. See Line 397-400 in the new version of manuscript. Raw reads were processed using fastp (v0.23.1) using a sliding window approach with a phred score quality threshold of 33 (–cut-right -q 33) with the --detect_adapter_for_pe parameter to remove adapter sequences, trim low-quality bases, and filter out poor-quality reads.
Comments 7: Usually, the default parameter for the log2foldchange is > 1, which is that at least 2 times a gene is expressed. The authors used 0.585, why do the authors used this cut-off value?
Response 7: Thank you for this important comment. We chose a log2 fold change cutoff of > 0.585, which corresponds to a fold change of 1.5, instead of the more stringent cutoff of > 1 (2-fold change), in order to capture a broader range of differentially expressed genes. This less stringent threshold has been widely used in transcriptomic studies to avoid overlooking genes with moderate but biologically relevant expression changes. Similar cut-off values have also been adopted in previous transcriptomic studies (listed below). The new references were added in the manuscript in Line 201.
- Song, X.; Zhang, M.; Wang, T.; Duan, Y.; Ren, J.; Gao, H.; Fan, Y.; Xia, Q.; Cao, H.; Xie, K.; et al. Polyploidization Leads to Salt Stress Resilience via Ethylene Signaling in Citrus Plants. New Phytologist 2025, 246, 176–191.
- Wang, Q.; Chen, H.; Du, B.; Wang, L.; Jing, P.; Wu, H.; Lin, J.; Gao, Y. A Human Tissue Map of 5-Hydroxymethylcytosines Exhibits Tissue Specificity through lncRNA Genes. Genomics 2025, 111085.
Comments 8: Please include the name of the softwares or the packages in R used to create the images.
Response 8: Thank you for your suggestion. We have clearly indicated the software version used. See Line 415-416, Differential expressed genes (DEGs) were identified using the DESeq2 package (v1.28.1) in R (v4.3.2). Line 419-420, PCA plot were then plotted with ggplot2 package in R (v4.3.2).
Reviewer 3 Report
Comments and Suggestions for Authors
This article ”Metabolomic and Transcriptomic Analyses provide Insights into Metabolic Networks during Citrus Fruit Development and Ripening” is well written, but here are some suggestions that would improve the quality of the manuscript
General
Please could you mention and emphasise some application-related findings of the study in the study objectives. The conclusion section needs to be improved. What about pigments, especially chlorophylls and carotenoids during citrus development and their possible link to ABA biosynthesis?
Specific
Try to avoid repetition of words both in the title and in the key words.
All Latin plant names must be italicised, but also the author's name of all plant species mentioned, including for citrus genera.
Taxonomic positions such as the abbreviations sp, spp. should not be italicised.
Line 83, 354. what does six stages of development mean, you probably mentioned some specific points during fruit development. I would suggest a different terminology, as there are three main stages of development. Before analysing, please explain why you chose these dates during citrus fruit development. Which three main stages of development are these specific points associated with.
Line 89 “The size of ‘kiyomi’ fruits increased as the fruit development stage”. Please, correct or change this sentence.
Why did you use the variety 'kiyomi' in the study? Explein the significance.
Please, provide the GPS coordinates of the orchard location
Here we need more information about the sample preparation before the analysis as well as the names of the equipment, and software you used (country and city of the producer)
It is not entirely clear which part of the fruit you take for all analyses in one count (exocarp, mesocarp or endocarp). In line 361 you use receptacle tissue. Which part of the fruit do you use that comes from flower receptacle? Sometimes you use the term pulp.
Comments on the Quality of English LanguageThe English could be improved.
Author Response
Dear Reviewer,
Thank you very much for your helpful comments. We have made the necessary revisions to the manuscript and have provided detailed responses below to clarify the points you raised. We believe these changes significantly improve the clarity and quality of our work.
Fang Song
2025.8.27
Point-by-point response to Comments and Suggestions for Authors
Comments 1: Please could you mention and emphasise some application-related findings of the study in the study objectives. The conclusion section needs to be improved. What about pigments, especially chlorophylls and carotenoids during citrus development and their possible link to ABA biosynthesis?
Response 1: Thanks a lot for your kind comments.
The application of this study is added in the introduction section in Line 90-93. “After that, the integrative analyses of metabolome and transcriptome data provided comprehensive information on the dynamics of major metabolites and the underlying regulatory pathways, which would further guide the breeding and regulation of shelf life in citrus.”
The conclusion section is revised as “The results suggest the potential involvement of ABA and specific transcription factors in mediating ripening-associated metabolic transitions and offer promising targets for genetic improvement and postharvest strategies.” See Line 446-448.
Previous studies on citrus fruit color have hypothesized that ABA may promote citrus peel coloration by affecting chlorophyll degradation and carotenoid accumulation. However, the tissue used in this study is citrus juice sacs, which contain almost no chlorophyll. The carotenoid accumulation mechanism is differed between juice sacs and peels. Therefore, this study does not focus on this topic.
Zhu, K.; Chen, H.; Mei, X.; Lu, S.; Xie, H.; Liu, J.; Chai, L.; Xu, Q.; Wurtzel, E.T.; Ye, J.; et al. Transcription Factor CsMADS3 Coordinately Regulates Chlorophyll and Carotenoid Pools in Citrus Hesperidium. Plant Physiology 2023, 193, 519-536.
Comments 2: Try to avoid repetition of words both in the title and in the key words.
Response 2: Thank you for this suggestion. We have revised the keywords to avoid repetition with the title. The updated keywords are: Transcriptome; Metabolome; Citric acid; Sucrose; Abscisic acid.
Comments 3: All Latin plant names must be italicised, but also the author's name of all plant species mentioned, including for citrus genera.
Response 3: Thank you for your suggestion. We have carefully revised the manuscript to ensure that all Latin plant names are consistently italicized. For example, Line 70, CitZAT5 (zinc finger of arabidopsis thaliana 5).
Comments 4: Taxonomic positions such as the abbreviations sp, spp. should not be italicised.
Response 4: Thank you for your helpful suggestion. We have revised the manuscript to ensure that taxonomic abbreviations such as sp. and spp. are not italicized. For example, Line 68, (Actinidia spp.).
Comments 5: Line 83, 354. what does six stages of development mean, you probably mentioned some specific points during fruit development. I would suggest a different terminology, as there are three main stages of development. Before analysing, please explain why you chose these dates during citrus fruit development. Which three main stages of development are these specific points associated with.
Response 5: Thank you for this valuable comment. Citrus fruit development is generally divided into three main stages: cell division (0-60 DAF), cell expansion (60-90 DAF), and fruit ripening (90-180 DAF). Since our study primarily focused on the dynamic changes in fruit metabolism, we selected six representative time points from 60 to 210 DAF (60, 90, 120, 150, 180, and 210 DAF) to cover the transition from the cell expansion stage through the fruit ripening stage, with 210 DAF representing the fully ripened stage.
We have revised the text accordingly: at line 87-90: “The metabolomes and transcriptomes were analyzed at six representative time points (60, 90, 120, 150, 180, and 210 days after flowering, DAF) covering the cell expansion (60-90 DAF) and ripening stages (90-180 DAF) of citrus fruit development.” and at line 361-364: “Fruits were randomly sampled at six developmental stages: 60, 90, 120, 150, 180, and 210 days after flowering (DAF) , corresponding to the cell expansion stage (60-90 DAF) and the fruit ripening stage (90-180 DAF). Juice sacs (pulp) were collected for subsequent analyses.”.
Comments 6: Line 89 “The size of ‘kiyomi’ fruits increased as the fruit development stage”. Please, correct or change this sentence.
Response 6: Thank you for pointing this out. We have corrected the sentence at line 89 to improve clarity, which now reads: “The size of ‘Kiyomi’ fruits increased progressively during fruit development (Fig. 1a).”
Comments 7: Why did you use the variety ‘kiyomi’ in the study? Explain the significance.
Response 7: Thank you for this important question. We selected the citrus variety ‘Kiyomi’ (Citrus unshiu × C. sinensis) for this study because it is a widely cultivated tangor hybrid with high commercial value due to its desirable flavor, characterized by a balanced ratio of sugars to organic acids. Moreover, ‘Kiyomi’ serves as an important genetic resource for citrus breeding programs and provides a representative model for studying sugar and organic acid metabolism in non-climacteric citrus fruits. Its well-defined genetic background and distinct flavor profile make it particularly suitable for dissecting the molecular and metabolic mechanisms underlying fruit ripening.
Comments 8: Please, provide the GPS coordinates of the orchard location.
Response 8: Thank you for your suggestion. We have added the GPS coordinates of the orchard location in the manuscript: (30.484075°N, 114.320471°E).
Comments 9: Here we need more information about the sample preparation before the analysis as well as the names of the equipment, and software you used (country and city of the producer).
Response 9: Thank you for this valuable comment. We have added more details on sample pretreatment in the Methods section and specified the names of the equipment and software used, including the manufacturer along with the country and city of origin. Line 383, Shimadzu, Kyoto, Japan; Applied Biosystems, Foster City, CA, USA. Line 392-395, the extracts were analyzed using a UPLC-ESI-MS/MS system (Shimadzu, Kyoto, Japan; Applied Biosystems, Foster City, CA, USA). Metabolite separation was followed by electrospray ionization tandem mass spectrometry (ESI-MS/MS, Applied Biosystems, Foster City, CA, USA). Line 405-406, HiPure HP Plant RNA Mini Kit (Magen, Guangzhou, China).
Comments 10: It is not entirely clear which part of the fruit you take for all analyses in one count (exocarp, mesocarp or endocarp). In line 361 you use receptacle tissue. Which part of the fruit do you use that comes from flower receptacle? Sometimes you use the term pulp.
Response 10: Thank you for pointing this out. For clarification, all samples used for analysis were taken from the juice sacs (pulp) of citrus fruits. We have revised the Methods section in lines 403-405: “Fruits were randomly sampled at six developmental stages (60, 90, 120, 150, 180, and 210 days after flowering, DAF), and juice sacs (pulp) were collected for subsequent analyses.” This revision explicitly specifies the sampled tissue.